# Puerarin Promotes the Migration and Differentiation of Myoblasts by Activating the FAK and PI3K/AKT Signaling Pathways

**DOI:** 10.3390/biology14010102

**Published:** 2025-01-20

**Authors:** Xiaofeng Fang, Hangjia Xu, Zhaoxin Fan, Hongge Yang, Yan Huang, Lin Xu, Yiwei Rong, Wei Ma, Liubao Pei, Hongsheng Liang

**Affiliations:** 1NHC Key Laboratory of Cell Transplantation, The First Affiliated Hospital of Harbin Medical University, Harbin 150001, China; 202101229@hrbmu.edu.cn (X.F.); xuhangjia630@163.com (H.X.);; 2Department of Orthopedics, The First Affiliated Hospital of Harbin Medical University, Harbin 150001, China

**Keywords:** puerarin, myoblasts, migration, differentiation, muscle regeneration

## Abstract

This study investigated the effects of puerarin, a flavonoid from the roots of radix *puerariae*, on skeletal muscle regeneration. By employing C2C12 myoblasts and BaCl_2_-based muscle injury models, we found that puerarin enhanced myoblast migration and differentiation by activating the FAK and PI3K/AKT signaling pathways. Puerarin also improved muscle regeneration after injury. These findings suggest that puerarin could be a promising drug candidate for treating muscle loss diseases.

## 1. Introduction

Skeletal muscle is one of the most abundant tissues in the human body and plays an important role in physical activities, metabolic balance, and temperature regulation [1]. The decline in muscle mass often occurs with trauma, aging, and various chronic pathological conditions (e.g., denervation, spinal cord injury, and cancers) [2,3,4]. Muscle loss impairs motor ability and increases the risk of complications such as osteoporosis and hyperglycemia, leading to increased incidence of disability and mortality [1,2,5]. However, the current therapies for such diseases are still insufficient.

It is well known that skeletal muscle possesses the capacity of regeneration. Satellite cells (SCs), residing between the plasma lemma and the basal lamina, are the main mediators of regeneration [6]. When stimulated by damage signals, these cells can re-enter the cell cycle to proliferate and then translate into myoblasts. Subsequently, the myoblasts migrate to the damage site, under the control of myogenic regulatory factors (MRFs) such as MyoD and MyoG; fuse with each other to form multinucleated myotubes; and finally differentiate into mature myofibers to reconstitute muscle tissues [7,8,9]. Muscle loss diseases have been shown to benefit from methods that promote myogenesis [10,11,12,13]. Therefore, researchers have been working to identify effective ways to stimulate SCs to promote muscle regeneration.

Puerarin is a flavonoid compound extracted from the root of radix *puerariae*. Its diverse biological functions, such as anti-inflammatory, antioxidant, antimicrobial, neuroprotective, and cardioprotective properties, have been widely studied [14]. Recently, emerging studies have reported that puerarin exerts positive physiological effects, such as proliferation and differentiation, on various stem cell populations, including neural progenitor cells and bone marrow mesenchymal stem cells [15,16,17]. However, whether puerarin can stimulate muscle stem cells to promote myogenesis remains unknown. In this study, we attempt to employ C2C12 myoblasts and the BaCl_2_-based muscle injury model to investigate the effect of puerarin on muscle regeneration and its underlying mechanism.

## 2. Materials and Methods

### 2.1. Cell Culture and Differentiation

Mouse C2C12 myoblasts (No. CL-0044) purchased from Procell (Wuhan, China) were maintained in high-glucose Dulbecco modified Eagle medium (DMEM; Gibco, Carlsbad, CA, USA) supplemented with 10% fetal bovine serum (FBS; Biological Industries, Kibbutz Beit-Haemek, Israel) and 1% penicillin–streptomycin (P/S; Solarbio, Beijing, China). When cells reached approximately 90% confluence, the medium was changed to differentiation medium (DMEM containing 2% horse serum (HS; Biological Industries, Kibbutz Beit-Haemek, Israel) and 1% PS) to induce differentiation for 4 days. The medium was replaced daily.

### 2.2. Cell Proliferation Assay

For the 5-Ethynyl-2′-deoxyuridine (EDU) assay, C2C12 myoblasts were seeded in 24-well plates at a density of 15,000 cells per well and then exposed to varying concentrations of puerarin (0, 5, 10, 20, 40, and 100 μM) for 24 h. Afterwards, the cells were cultured in 20 mM EDU medium (Beyotime, Shanghai, China) for 2 h, followed by fixation with 4% paraformaldehyde and permeabilization with 0.5% Triton X-100. The cells were subsequently stained with Click solution and DAPI solution. Images were finally acquired by a fluorescence microscope (Zeiss, Jena, Germany), and Image J software (64-bit Java 1.8.0_172) was employed to analyze EDU-positive cell numbers.

For the Cell Counting Kit-8 (CCK-8) Assay, the cells were incubated in 96-well plates at a density of 3000 cells per well. After drug treatment, 10 μL of CCK8 solution (Abbkine, Wuhan, China) was added to each well; then, the plate was incubated at 37 °C for 1 h. Finally, the optical density value at 450 nm was quantified by using an enzyme mark instrument (Biotek, Winooski, VT, USA).

### 2.3. Scratch Wound Assay

C2C12 myoblasts were placed in 6-well plates at a concentration of 1.0 × 10^6^ cells per well. Upon reaching 90% cell confluence, a sterile 200 μL pipette tip was employed to scratch cells at the center of the well, and the cellular debris was cleared with PBS. DMEM containing puerarin (0, 5, 10, 20, and 40 μM) was added for incubation, and the FAK inhibitor PF-573228 was added at a concentration of 10 μM if required [18]. We photographed the same wound place with a microscope (OLYMPUS, Tokyo, Japan) at 0 and 24 h after scratching. The migration ratio was calculated by the percentage of healed wound area, and the area of the scratch was quantified by using Image J software (64-bit Java 1.8.0_172).

### 2.4. Transwell Migration Assay

C2C12 myoblasts (2 × 10^4^ cells/well) were inserted into the upper transwell chamber (8 µm pore size) of 24-well plates and exposed to FBS-free DMEM. Then, 750 µL of complete DMEM with/without drugs and inhibitor was added to the lower chamber. After 24 h, the un-migrated myoblasts were wiped away with a cotton swab, while the migrated cells were fixed by 4% paraformaldehyde and colored with 0.5% crystal violet. The migration of myoblasts was photographed through a microscope (Zeiss, Jena, Germany). The crystal violet that had stained the migrated cells was extracted with 0.1% acetic acid, and the cell migration capacity was quantified by the optical density of crystal violet at 570 nm [19].

### 2.5. Western Blot Analysis

The C2C12 cells or muscle tissues were mixed with proteinlysis buffer (Beyotime, Shanghai, China) containing protease/phosphatase inhibitors, and the proteins were isolated through a centrifugation process at 14,000 rpm for 15 min at 4 °C. Then, equitable amounts of protein samples were resolved by electrophoresis using 7.5% or 10% polyacrylamide gel and transferred onto a polyvinylidene fluoride membrane (Merck Millipore, Darmstadt, Germany). The membranes were treated with blocking solution (5% skim milk or bovine serum albumin) and immersed with primary antibodies for 12–16 h at 4 °C. Subsequently, the membranes were rinsed by tris-buffered saline with Tween 20 and then incubated with the secondary antibodies for 1 h at 37 °C. After another round of washing, the protein blots were developed by ECL reagent (Abbkine, Wuhan, China) and visualized through the chemiluminescence imager system (Syngene, Cambridge, UK). The primary antibodies MHC (1:1000; Proteintech, Wuhan, China), MyoD (1:1000; Proteintech), MyoG (1:1000; ImmunoWay, Plano, TX, USA), Myf5 (1:1000; ImmunoWay), p-PI3K (1:1000; Bioss, Beijing, China), PI3K (1:1000; Bioss), p-AKT (1:1000; Proteintech), AKT (1:5000; Proteintech), p-FAK (1:1000; Bioss), and FAK (1:5000; Proteintech) were employed. The secondary antibodies conjugated with HRP were acquired from Abbkine.

### 2.6. Immunofluorescence Analysis

Following 4 days of differentiation, C2C12 myotubes in 6-well plates were fixed with 4% paraformaldehyde for 20 min and then permeabilized with 0.5% Triton X-100 in PBS for 15 min. After that, blocking was performed with PBSTx buffer containing 5% BSA for 60 min at 37 °C. Then, the samples were incubated with the primary antibody MHC (1:400; Bioss) overnight at 4 °C and subsequently with Alexa Fluor 488-conjugated secondary antibody (1:500; Abbkine) for 1 h at 37 °C. The nuclei were counter-stained by DAPI. At least 3 randomly selected areas from each well were captured by a fluorescence microscope (Zeiss, Germany), and the fusion index (the ratio of nuclei within myotubes to the overall nuclear count) was analyzed by Image J software (64-bit Java 1.8.0_172).

### 2.7. Animals and Skeletal Muscle Injury Model

The vivo experiment was conducted according to the guidelines of the Declaration of Helsinki and was approved by the Institutional Animal Care and Use Committee of The First Clinical Medical College of Harbin Medical University (No. YS078). Eight-week-old male C57 mice purchased from Changsheng biotechnology Co., Ltd. (Changchun, China) were employed in this study. To establish a muscle injury model, we injected 50 µL of physiological saline into the tibialis anterior muscle of mice in the non-injury group and 50 µL of a 1.2% barium chloride solution (prepared by dissolving 0.12 g of barium chloride in 10 mL of saline) into the tibialis anterior muscle of mice in the injury group [20]. To assess the positive effect of puerarin on muscle regeneration, the drugs were given orally at a daily dose of 100 mg per kilogram of body weight. According to previous research, this dose is considered safe and beneficial to skeletal muscle [21]. TA muscles were harvested 5 days post-injury.

### 2.8. Hematoxylin and Eosin (H&E) Staining

TA muscle samples were collected, fixed in a 4% paraformaldehyde solution for 24 h, and then embedded in paraffin. After being prepared as paraffin sections, H&E staining was performed based on the protocol of the manufacturer (Beyotime, Shanghai, China). Images were captured by a Zeiss microscope.

### 2.9. Statistical Analysis

Data were collected from at least three independent experiments and expressed as the means ± SDs. The analysis of statistical significance between two groups was performed by using Student’s *t*-test, while for multiple groups, one-way ANOVA analysis was employed. Differences were regarded as significant at *p* < 0.05. All statistical tests were performed by using GraphPad Prism software 8.0.2 (GraphPad Software Inc., La Jolla, CA, USA).

## 3. Results

### 3.1. The Proliferative Effect of Puerarin on C2C12 Myoblasts

To estimate the influence of puerarin on the growth capacity of C2C12 cells, puerarin at concentrations ranging from 0 to 100 µM was employed to treat the cells. After 24 h of exposure to varying concentrations of puerarin, the proportion of EDU-positive myoblasts in each treatment group at concentrations below 100 µM was approximately 59%, with no significant difference from the control. However, at 100 µM, the proportion of EDU-positive cells markedly decreased to 48% (Figure 1A,B). Consistently, the CCK8 assay demonstrated that cell absorbance significantly decreased at a puerarin concentration of 100 μM (Figure 1C). Thus, puerarin did not promote proliferation in C2C12 cells, and subsequent experiments should consider concentrations below 100 μM to ensure non-toxicity.

### 3.2. Puerarin Promoted the Migration of C2C12 Myoblasts

We conducted 24 h scratch wound and transwell migration assays to assess the effect that puerarin might have on myoblast migration. As shown in Figure 2A,B, the migration rates of C2C12 cells treated with 10, 20, and 40 μM puerarin were 62.82 ± 3.69%, 66.85 ± 3.90%, and 65.13 ± 2.78%, respectively, which were significantly higher than that of the control group (48.19 ± 2.24%). Moreover, in the transwell model, the number of C2C12 myoblasts that migrated through the membrane significantly increased with treatment with 10, 20, and 40 μM puerarin (Figure 2C,D). These outcomes demonstrated that puerarin could enhance the migratory capacity of C2C12 myoblasts.

### 3.3. Puerarin-Induced Myoblast Migration Through FAK Signaling

FAK signaling has been demonstrated to play a crucial role in myoblast migration. Accordingly, we sought to determine if the puerarin-induced migration in C2C12 cells was mediated by the FAK pathway. Considering that the myoblasts exhibited the strongest migratory ability at a concentration of 20 µM puerarin, we treated the cells with puerarin at concentrations of 0 and 20 µM during the migration process. We performed a Western blot test and found that puerarin (20 μM) could increase the phosphorylation of FAK protein (Figure 3A,B). Subsequently, we utilized the FAK inhibitor PF-573228 to suppress the activation of FAK signaling (Figure 3C,D). We concurrently observed that PF-573228 effectively neutralized the migratory enhancement induced by puerarin in the myoblasts across the scratch and transwell assays (Figure 3E–H). These results proved that puerarin promoted myoblast migration through the FAK pathway.

### 3.4. Puerarin Stimulated the Differentiation of C2C12 Myoblasts

To study the effect of puerarin on the differentiation of C2C12 cells, we added puerarin at various concentrations (0 to 40 μM) to a 2% differentiation medium to induce myoblast differentiation for 4 days. Under a light microscope, we noted a marked enhancement in myotube formation in the puerarin (10, 20, and 10 μM)-treated group as opposed to the control group (Figure 4A). By using MyHC immunofluorescence staining, we also observed a significant increase in the fusion index of myoblasts in the puerarin (10, 20, and 40 μM)-treated groups, with the 20 μM puerarin group showing the highest increase (Figure 4B,C). Subsequently, we evaluated the changes in the expression levels of differentiation-related proteins after 20 μM puerarin treatment, and as expected, puerarin increased the expression of the muscle-related proteins MHC, MyoD, and MyoG (Figure 4D,E). These results demonstrated that puerarin could promote the myogenic differentiation of C2C12 cells.

### 3.5. Puerarin-Induced Myoblast Differentiation Through PI3K/AKT Pathway

To investigate the potential intracellular mechanisms involved in puerarin-induced myoblast differentiation, the expression of proteins associated with signaling pathways were determined by using Western blot analysis. Our findings revealed that puerarin notably elevated the ratios of p-PI3K/PI3K and p-AKT/AKT, suggesting the activation of the PI3K/AKT signaling pathway (Figure 5A,B). We further utilized the PI3K inhibitor LY294002 and found that puerarin treatment did not increase the phosphorylation of PI3K and AKT after inhibitor treatment (Figure 5C,D). Moreover, we also found that after treatment with LY294002, puerarin treatment did not lead to an increase in the fusion rate of myoblasts and the expression levels of the MHC, MYOD, and MYOG proteins (Figure 5E,I). Taken together, puerarin promotes muscle differentiation via the PI3K/AKT signaling pathway.

### 3.6. Puerarin Promotes Muscle Regeneration After Muscle Damage

To further assess the influence of puerarin on muscle regeneration in vivo, a model of TA muscle injury was created through the injection of BaCl_2_. We treated mice with puerarin (100 mg/kg/day) for 5 days following muscle injury. H&E staining revealed that the damaged muscles in puerarin-treated mice generated a significantly greater quantity of newly formed muscle fibers than those in the control group (Figure 6A,B). Consistently, the expression of MyHC proteins was found to be upregulated in the puerarin-treated TA muscles (Figure 6C,D). These studies indicated that puerarin could facilitate the repair of muscle injury by promoting muscle regeneration.

## 4. Discussion

The loss of skeletal muscle is gradually becoming a severe social problem, especially with the advent of an aging population [2]. Regulating myogenesis is considered to be an important means for muscle mass maintenance. However, there are still no reliable strategies to promote muscle regeneration in clinical settings [22]. Flavonoid puerarin is the principal active constituent extracted from the root of radix *puerariae*, which has been approved by the China Food and Drug Administration to treat ischemic cardiomyopathy and hypertension [23]. Accumulating evidence suggests that puerarin can function as a modulator of tissue regeneration, and its positive effects on osteanagenesis and neurogenesis have been proved [15,17]. However, whether pueraria promotes myogenesis remains unknown. In this study, we utilized C2C12 myoblasts and the BaCl_2_-based muscle injury model to assess the effects of puerarin on muscle regeneration. We found that puerarin could promote myogenesis by stimulating myoblast migration and differentiation.

In the process of muscle regeneration, stem cells need to engage in symmetric divisions first to increase their numbers upon activation. Puerarin has been recognized as a potent stimulator of stem cell proliferation. It is reported that puerarin at concentrations of 10 µM and 20 µM substantially enhanced the proliferative capacity of bone marrow stromal cells and osteoblasts [24,25]. Furthermore, recent research indicated that 10 µM puerarin significantly promoted the growth of periodontal ligament stem cells and nucleus pulposus mesenchymal stem cells [26,27]. In present work, we treated myoblasts with similar concentration gradients of puerarin, but no markedly higher proliferation activity was observed in cells exposed to the drug. This result could be attributed to the inherent specificity of the cell types involved. Thus, puerarin induced myogenesis without influencing the growth capacity of stem cells.

Following proliferation, mononuclear myoblasts need to move towards one another or pre-existing myotubes to form longer and thicker myotubes [6,9,22]. Our wound healing and transwell assays indicated that puerarin could augment the migration of myoblasts. Currently, the modulatory effect of the FAK/Paxillin pathway on myoblast migration has been emphasized. It has been reported that the different migration capacities of myoblast are related to differences in the localization and distribution of F-actin and focal adhesion, which are modulated by FAK signaling [18,28,29,30,31]. Hour et al.’s study found that flavonoid quercetin promoted the migration of C2C12 cells with increased expression of phosphorylated FAK [32]. In this study, we also discovered upregulated levels of phosphorylated FAK after puerarin treatment, and this effect could be inhibited by the FAK inhibitor PF-573228. Moreover, PF-573228 concurrently eliminated the increased motility of myoblasts induced by puerarin in the wound healing and transwell assays. These results strongly suggest that puerarin promoted C2C12 myoblast migration through the FAK pathway. Notably, PF-573228 is capable of inhibiting cell migration in a dose-dependent manner [18,33]. In this study, we selected a higher concentration of 10 µM as reported in the literature, which consequently demonstrated a potent inhibitory effect. We hypothesize that this may be the reason for the absence of puerarin’s pro-migratory effect in the presence of the inhibitor.

Terminal cell differentiation is the most critical step in myogenesis, where myoblasts merge into multinucleated myotubes under the regulation of MRFs with the expression of the muscle-specific protein MHC [34,35]. In this research study, we observed markedly higher rates of myotube fusion and levels of MHC and MRFs (MyoD and MyoG) in the puerarin-treated group, indicating that puerarin could stimulate myoblast differentiation. Abundant evidence has demonstrated that the PI3K/AKT pathway represents the major regulatory mechanism controlling myoblast fusion and myotube hypertrophy. Phosphorylated PI3K causes the activation of the serine/threonine kinase AKT, which subsequently activates the downstream molecules involving mTOR, p70S6K, 4EBP1, and eIF4E, ultimately leading to elevated muscle synthesis [36,37,38,39,40]. Notably, PI3K/AKT signaling has also been found to be a main mechanism through which puerarin facilitates both osteogenic and neuronal differentiation [15,17]. Here, we found a significant increase in the phosphorylation levels of the PI3K and AKT proteins. We then employed LY294002, a PI3K inhibitor, to ascertain the role of the PI3K/AKT pathway in puerarin-induced myoblast differentiation. According to the data, LY294002 suppressed the phosphorylation of PI3K and AKT induced by puerarin. Moreover, it also abolished the expression of myogenic differentiation markers (MHC and MRFs) stimulated by puerarin. These findings demonstrated that puerarin promoted myoblast differentiation through PI3K/AKT pathway. Our study also found that in the presence of the PI3K inhibitor LY294002, puerarin could not enhance differentiation activity, which is consistent with the findings of Yu-Kuei Chen et al., who showed that LY294002 could inhibit myogenic differentiation regardless of the presence or absence of quercetin [32]. Previously, Eric R. Blough et al. conducted early studies by using cells with targeted knockout or genetic deficiency, indicating that AKT is crucial for muscle differentiation [41,42]. Our results seem to further confirm the irreplaceable role of AKT activation in the differentiation of myoblasts.

Based on above in vitro studies, puerarin should be a stimulant for skeletal muscle regeneration. To further confirm the role of puerarin in myogenesis, we established a TA injury muscle model induced by BaCl_2_. Upon skeletal muscle injury, a program of repair and regeneration is initiated. It is reported that the histological characteristics of injury muscle vary over time, with acute inflammation on day 1, cell proliferation on day 3, and muscle fiber formation on days 5–7 [43,44]. Previous research has found that the myogenic stimulants catechins can significantly increase the number of large-diameter muscle fibers on day 5 post-muscle injury by activating satellite cells and promoting their differentiation [45]. Similarly, lithocholic acid has been shown to significantly increase the area of newly formed muscle fibers on day 5 post-muscle injury by promoting the proliferation and differentiation of satellite cells [46]. Here, we conducted H&E staining 5 days post-injury and found that puerarin increased the number of central nucleated myofibers [40]. Meanwhile, increased expression of MHC was observed in the puerarin-treated group, which was consistent with the cellular experiments. These data indicated that puerarin was capable of stimulating myogenesis within living organisms, improving the repair of muscle injuries. We believe these positive effects are closely related to its ability to promote the migration and differentiation of muscle stem cells.

## 5. Conclusions

In conclusion, our study demonstrated that puerarin promoted the migration and differentiation of C2C12 myoblasts via the FAK and PI3K/AKT pathways. Furthermore, puerarin treatment was found to enhance myogenesis in injured muscle. These findings suggest that puerarin participates in muscle regeneration with the activation of myoblast migration and differentiation. To our best knowledge, this is the first study to elucidate the role of puerarin in myogenesis. Our findings provide a new candidate drug for the treatment of muscle loss diseases.

## Figures and Tables

**Figure 1 biology-14-00102-f001:**
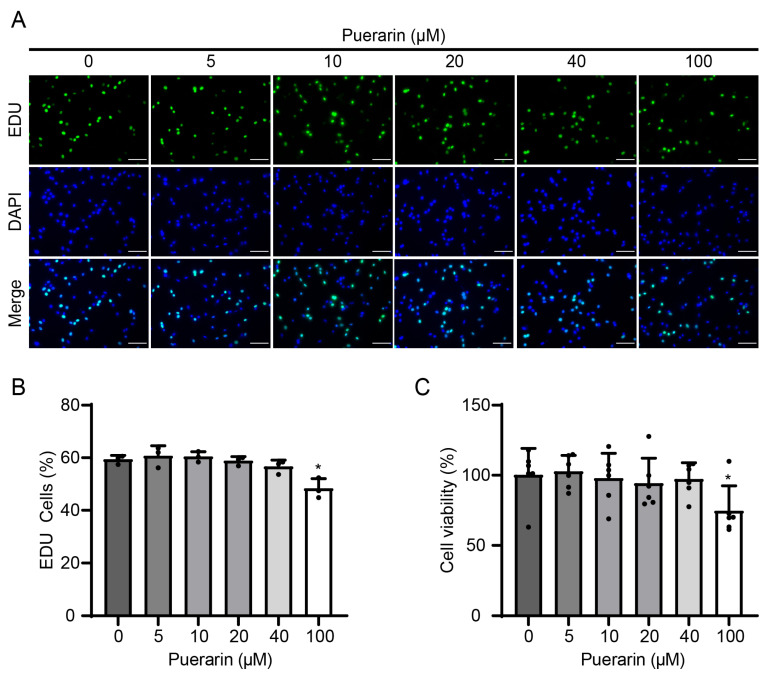
The proliferative effect of puerarin on C2C12 myoblasts. C2C12 cells were cultured in proliferation medium supplemented with puerarin (0, 5, 10, 20, 40, and 100 µM) for 24 h. (**A**) The EDU incorporation assay was employed to evaluate the cell proliferation ability. Scale bar = 100 µm. (**B**) Percentage of EDU-positive cells in panel (**A**). Data are displayed as the means ± SDs of three independent experiments. (**C**) Cell viability tested by the CCK8 assay. * *p* < 0.05 compared with the control group.

**Figure 2 biology-14-00102-f002:**
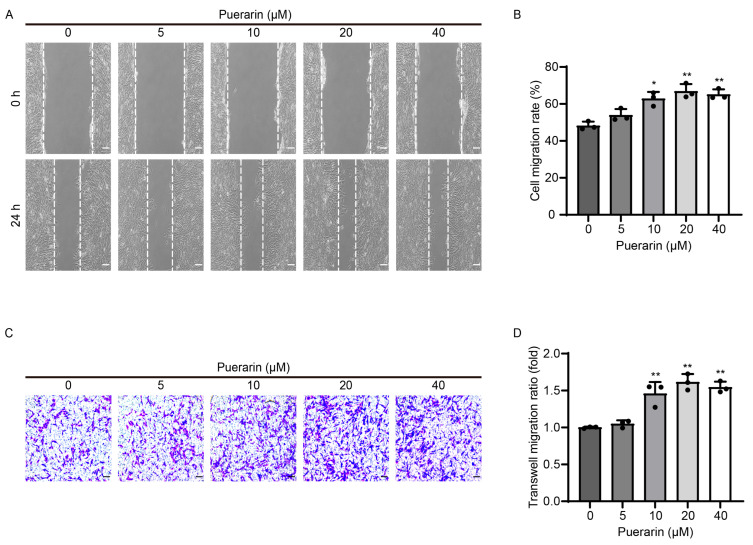
Puerarin promoted the migration of C2C12 myoblasts. C2C12 cells were incubated with DMEM for 24 h with varying concentrations of puerarin (0, 5, 10, 20, and 40 µM). (**A**) The wound healing assay was conducted to assess cell migration by introducing a scratch in a confluent monolayer and monitoring it for 24 h. Scale bar = 100 µm. (**B**) The data of cell migration ratio based on (**A**). (**C**) The transwell assay was conducted to measure the C2C12 cell movement capacity. Scale bar = 100 µm. (**D**) Quantitative analysis of crystal violet’s optical density at 570 nm in panel C. Data are displayed as the means ± SDs of three independent experiments. * *p* < 0.05 and ** *p* < 0.01 compared with the control group.

**Figure 3 biology-14-00102-f003:**
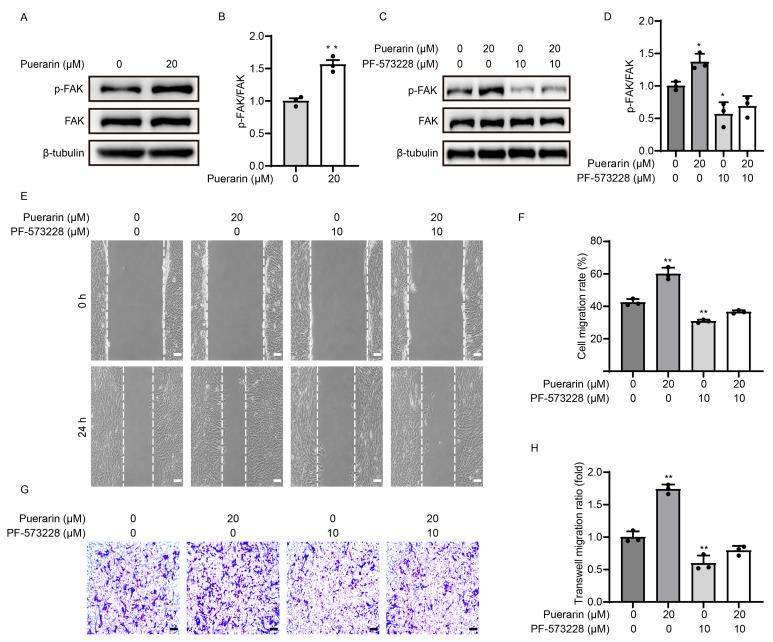
Puerarin-induced myoblast migration through FAK signaling. C2C12 myoblasts were treated with puerarin (20 µM), PF-573228 (10 µM), or both in DMEM for 24 h. (**A**,**C**) The protein expression of p-FAK (Tyr397) was analyzed by Western blot. β-Tubulin was employed as the loading control. (**B**,**D**) Quantitative analysis of protein expression of p-FAK (Tyr397). (**E**) The wound healing assay was performed to evaluate cell migration by making a scratch in a confluent monolayer and monitoring it for 24 h. Scale bar = 100 µm. (**F**) The migration ratio based on the scratch wound assay. (**G**) The transwell assay was performed to assess the migration ability of C2C12 cells. Scale bar = 100 µm. (**H**) Quantitative analysis of crystal violet’s optical density at 570 nm in panel (**G**). Data are displayed as the means ± SDs of three independent experiments. * *p* < 0.05 and ** *p* < 0.01 compared with the control group.

**Figure 4 biology-14-00102-f004:**
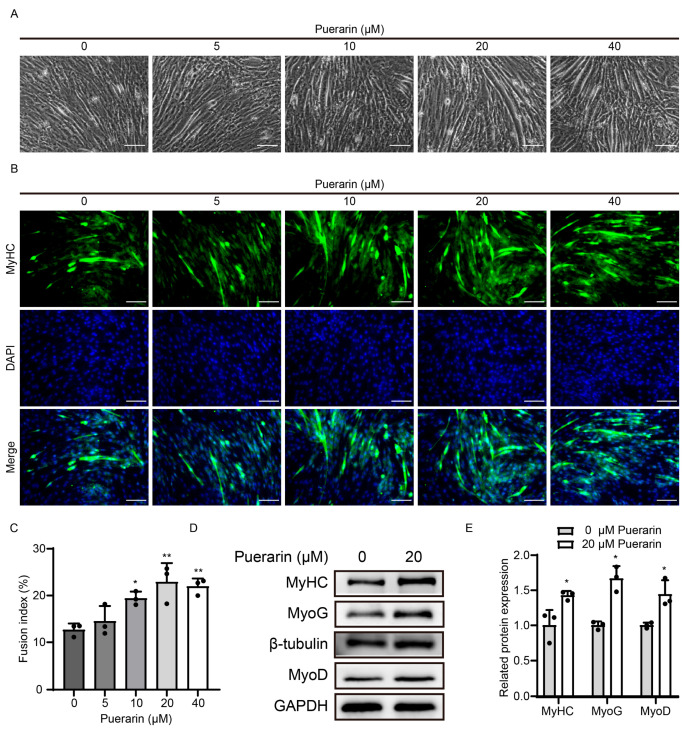
Puerarin stimulated the differentiation of C2C12 myoblasts. C2C12 cells were cultured in differentiation medium for 4 days with varying concentrations of puerarin (0, 5, 10, 20, and 40 µM). (**A**) Representative optical images showing myotubes after 4 days of differentiation. Scale bar = 50 µm. (**B**) Immunofluorescence staining of MyHC (green) and DAPI (blue) in myotubes. Scale bar = 100 µm. (**C**) Fusion index (the proportion of nuclei in cells expressing MHC) in figure (**B**). (**D**) The protein expression levels of MyHC, MyoD, and MyoG were analyzed by Western blot. β-Tubulin or GAPDH was used as the loading control. (**E**) Quantitative analysis of protein expression of MyHC, MyoD, and MyoG. Data are displayed as the means ± SDs of three independent experiments. * *p* < 0.05 and ** *p* < 0.01 compared with the control group.

**Figure 5 biology-14-00102-f005:**
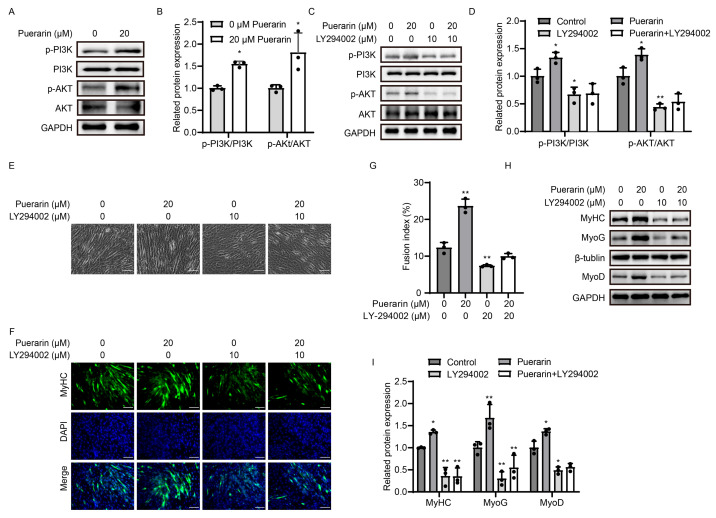
Puerarin-induced myoblast differentiation through the PI3K/AKT pathway. C2C12 myoblasts were treated with puerarin (20 µM), LY294002 (10 µM), or both in differentiation medium for 4 days. (**A**,**C**) The protein expression levels of p-PI3K (Tyr317) and p-AKT (T308) were analyzed by Western blot. GAPDH was employed as the loading control. (**B**,**D**) Quantitative analysis of protein expression of p-PI3K (Tyr317) and p-AKT (T308). (**E**) Representative optical images of myotubes following 4 days of differentiation. Scale bar = 50 µm. (**F**) Myotubes were stained for MHC (green) and DAPI (blue). Scale bar = 100 µm. (**G**) Fusion index (the proportion of nuclei in cells expressing MHC) in figure (**F**). (**H**) The protein levels of MyHC, MyoD, and MyoG were tested by Western blot. β-Tubulin or GAPDH was used as the loading control. (**I**) Quantitative analysis of protein levels of MyHC, MyoD, and MyoG. Data are displayed as the means ± SDs of three independent experiments. * *p* < 0.05 and ** *p* < 0.01 compared with the control group.

**Figure 6 biology-14-00102-f006:**
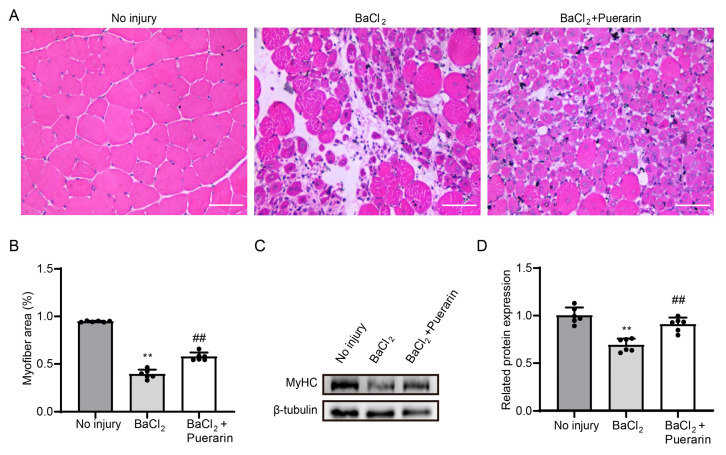
Puerarin promotes muscle regeneration after muscle damage. After injecting 1.2% BaCl_2_ in the middle of the TA muscle, the muscle was collected following continuous oral administration of puerarin (100 mg/kg/day) for 5 days. (**A**) H&E staining of the TA muscle 5 days post-injury. Scale bar = 50 µm. (**B**) Quantitative analysis of myofiber area in figure (**A**). (**C**) The protein expression of MyHC was analyzed by Western blot. β-Tubulin was used as the loading control. (**D**) Quantitative analysis of protein expression of MyHC. Data are displayed as the means ± SDs of six independent experiments. ** *p* < 0.01 compared with the control group. ## *p* < 0.01 compared with the injury group.

## Data Availability

The data presented in this study are available on request from the corresponding author.

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
