# Peer review of "Puerarin Promotes the Migration and Differentiation of Myoblasts by Activating the FAK and PI3K/AKT Signaling Pathways"

_biology, 2025, doi:10.3390/biology14010102_

Round 1

Reviewer 1 Report

Comments and Suggestions for Authors

The manuscript presents data on the effects of puerarin on C2C12 myoblast migration and differentiation, focusing on the FAK signaling pathway and PI3K/AKT pathway. While the study addresses an interesting question with potential implications for muscle repair, several issues need to be addressed for clarity, rigor, and scientific validity.

1. Concentration Testing: The authors only tested concentrations up to 40 µM in the migration assays. A rationale for this specific choice of concentration should be provided. Why was this upper limit chosen, and could additional concentrations yield more insight? 

The choice of 20 µM for assessing FAK signaling needs clarification. Why was this concentration chosen, and how does it relate to the concentrations used in migration assays?

2. The overall image quality in the figures is subpar, with the majority appearing small and of low resolution. Only Figure 6 exhibits clear images. Consider enhancing image quality so that all figures are visually comparable and interpretable.

3. When plotting data in graphs, all data points should be shown. This practice would provide a clearer representation of data variability and strengthen the overall presentation.

4. Figure Clarifications needed:

   Figure 1a - It should specify what is being shown rather than just mentioning the assay.

   Figure 2 - The scale bars are not clear. Consider showing a smaller portion of the image more clearly and keeping the complete images in the supplementary materials if needed.

   Figure 3 Legend - Requires corrections and to mention all the G and H images. The legend currently states, "(C) The cells' migratory capabilities were assessed by a wound healing assay. Scale bar = 100 μm." This should be revised for clarity. 

   Figure 5 - The manuscript does not explain why the addition of puerarin, in the presence of the inhibitor LY294002, did not lead to an increase in activity. This requires discussion. Could this be due to the same underlying mechanism? Any information on the direct target of puerarin would be valuable for understanding this observation.

5. Phosphorylation Notation: There is inconsistency in the notation of phosphorylated proteins. For instance, "phospho PI3K" is indicated as p-PI3K in text while shown as P-PI3K in images. Such discrepancies should be addressed across all figures for all proteins to maintain consistency.

Specific Discussion Points:

6. The manuscript states on line 313 that "these results strongly suggested puerarin promoted C2C12 myoblasts migration through the FAK pathway." However, it then indicates on line 331 that puerarin abolished the expression of myogenic differentiation markers stimulated by puerarin. Why is there no discussion around the lack of rescue effects of puerarin in the presence of the inhibitor?

7. Regarding in vivo studies, was the control group subjected to saline injection? Comparing the injured site to a non-injured site might not provide enough control for the experiment's reliability. Consider including controls with saline-injected tibialis anterior (TA) muscles.

8. In the discussion, the assertion that muscle repair is facilitated by puerarin is overly simplistic. Exploring other injury models and contrasting the findings could provide greater insights, particularly regarding the mechanistic roles involved.

Minor Typo errors:

9. Some minor English language mistakes are present throughout the manuscript, particularly in the use of terms. For instance, “Favonoid” in line 281 should be corrected to “Flavonoid.” In figure 6B the “2” in BaCl2 appears cut off. Consider replacing ‘uninjury’ with ‘no injury’ or simply say control and mention it in the legend that the control is no injury sample.

In conclusion, the manuscript presents a potentially valuable contribution to the field; however, substantial revisions are needed to improve clarity, provide deeper analysis, and ensure scientific rigor. I recommend reevaluating the figures, enhancing data presentation, and expanding the discussion to adequately address the outlined concerns.

Author Response

Response to Reviewer 1 Comments

1. Summary

We appreciate you very much for the positive comments and constructive advice on our manuscript entitled “Puerarin promotes the migration and differentiation of my-oblasts by FAK and PI3K/AKT signaling pathways”. (Ref.: biology-3345306). We did do our best to revise our manuscript according to the comments. We have already made a response marked in red font. We would like to express our great appreciation to you.

2. Point-by-point response to Comments and Suggestions for Authors

Comments 1: Concentration Testing: The authors only tested concentrations up to 40 µM in the migration assays. A rationale for this specific choice of concentration should be provided. Why was this upper limit chosen, and could additional concentrations yield more insight? The choice of 20 µM for assessing FAK signaling needs clarification. Why was this concentration chosen, and how does it relate to the concentrations used in migration assays?

Response 1: Thank you very much for your question. We initially established a concentration gradient of Puerarin from 0 to 100 µM to investigate the effect of Puerarin on the proliferation of myoblasts. The reason for choosing this concentration range has been mentioned in the discussion (line 293-299). However, we observed that when the concentration reached 100 µM, cell proliferation was inhibited. To exclude this effect, we used the remaining known safe concentration range (up to 40 µM) in our migration analysis. Additional concentrations might yield further insights, and expanding the concentration range could be considered in future studies to further explore potential migration effects at higher concentrations. However, in the current experiment, concentrations from 0 to 40 µM have provided sufficient data to elucidate the positive effect of puerarin on the migration ability of myoblasts. Additionally, since the cells exhibited the strongest migratory ability at a concentration of 20 µM puerarin, we selected this concentration to assess the impact of the FAK signaling on cell migration. We have added clarification in the text (line 193-195). Thanks for your kind suggestion again.

Comments 2: The overall image quality in the figures is subpar, with the majority appearing small and of low resolution. Only Figure 6 exhibits clear images. Consider enhancing image quality so that all figures are visually comparable and interpretable.

Response 2: Thank you very much for your suggestions. We sincerely apologize for the small size and low resolution of our images. We have updated all images to higher resolution versions (600 dpi) and have also adjusted their dimensions to ensure optimal clarity and visibility.

Comments 3: When plotting data in graphs, all data points should be shown. This practice would provide a clearer representation of data variability and strengthen the overall presentation.

Response 3: Thank you very much for your suggestion. We have added data points to all graphs to clearly represent the variability of the data.

Comments 4:

Figure Clarifications needed:

Figure 1a - It should specify what is being shown rather than just mentioning the assay. Figure 2 - The scale bars are not clear. Consider showing a smaller portion of the image more clearly and keeping the complete images in the supplementary materials if needed.

Figure 3 Legend - Requires corrections and to mention all the G and H images. The legend currently states, "(C) The cells' migratory capabilities were assessed by a wound healing assay. Scale bar = 100 μm." This should be revised for clarity.

Figure 5 - The manuscript does not explain why the addition of puerarin, in the presence of the inhibitor LY294002, did not lead to an increase in activity. This requires discussion. Could this be due to the same underlying mechanism? Any information on the direct target of puerarin would be valuable for understanding this observation.

Response 4: Thank you very much for your suggestion. For Figure 1a, we have added a clearer description (line 157-161) to specify what is being shown, ensuring that it is more informative than just mentioning the assay. In response to the issue with Figure 2, we have revised the figure to include clearer scale bars. Regarding Figure 3, we have corrected the legend to include all relevant images (G and H) (line 211-212) and have revised the statement you pointed out to improve clarity (line 208-210). For Figure 5, we have added a discussion to explain why the addition of puerarin in the presence of the inhibitor LY294002 did not lead to an increase in activity (line 339-345). We believe that the "determining" role of AKT protein in myoblast differentiation and the inhibition of AKT by LY294002 are the reasons for this phenomenon.

Comments 5: Phosphorylation Notation: There is inconsistency in the notation of phosphorylated proteins. For instance, "phospho PI3K" is indicated as p-PI3K in text while shown as P-PI3K in images. Such discrepancies should be addressed across all figures for all proteins to maintain consistency.

Response 5: Thank you for pointing out the error. We have corrected the notation of phosphorylated proteins in the figures to maintain consistency throughout the text, as can be seen in figures 3 (Page 6) and figures 5 (Page 8).

Comments 6: The manuscript states on line 313 that "these results strongly suggested puerarin promoted C2C12 myoblasts migration through the FAK pathway." However, it then indicates on line 331 that puerarin abolished the expression of myogenic differentiation markers stimulated by puerarin. Why is there no discussion around the lack of rescue effects of puerarin in the presence of the inhibitor?

Response 6: Thank you very much for your good question. According to our data, the effects of puerarin in promoting cell migration and differentiation were both inhibited in the presence of inhibitors (PF-573228 or LY294002). For the inhibition of cell migration, we believe it may be related to the higher concentrations of inhibitors we used, and PF-573228 can inhibit cell migration in a dose-dependent manner. As for the loss of differentiation effects, we think it is related to the decisive role of AKT protein in myoblast differentiation and the inhibitory effect of LY294002 on AKT. We have detailed these findings in the corresponding sections of the discussion (line 316-320) and (line 339-345).

Comments 7: Regarding in vivo studies, was the control group subjected to saline injection? Comparing the injured site to a non-injured site might not provide enough control for the experiment's reliability. Consider including controls with saline-injected tibialis anterior (TA) muscles.

Response 7: Thank you very much for your good question. We administered 50 µL of physiological saline into the tibialis anterior muscle of mice in the control group, and 50 µL of a 1.2% barium chloride solution (prepared by dissolving 0.12 g of barium chloride in 10 mL of saline) into the tibialis anterior muscle of mice in the injury group. We have added this information in the methods section. (line 135-138).

Comments 8: In the discussion, the assertion that muscle repair is facilitated by puerarin is overly simplistic. Exploring other injury models and contrasting the findings could provide greater insights, particularly regarding the mechanistic roles involved.

Response 8: Thank you for your suggestion. We acknowledge that our discussion regarding the role of puerarin in facilitating muscle repair was overly simplistic. Consequently, we have expanded our literature review to to analyze the different effects and mechanisms of other drugs on muscle injury models.(line 351-356) These additional analyses have helped to enhance the depth of our research. Thank you again for your advice.

Comments 9: Some minor English language mistakes are present throughout the manuscript, particularly in the use of terms. For instance, “Favonoid” in line 281 should be corrected to “Flavonoid.” In figure 6B the “2” in BaCl2 appears cut off. Consider replacing ‘uninjury’ with ‘no injury’ or simply say control and mention it in the legend that the control is no injury sample.

Response 9: Thank you very much for your kind reminder. We have made the corresponding correction to the language errors or inaccuracies in the manuscript and figures, as can be seen in line 283 and figures 6 (Page 8). "For Figure 6, we have replaced 'uninjury' with “no injury “. Thank you again for your correction and suggestion.

Reviewer 2 Report

Comments and Suggestions for Authors

The paper entitled “Puerarin promotes the migration and differentiation of myoblasts by FAK and PI3K/AKT signaling pathways” by Fang et al. is investigating the role of puerarin on skeletal muscle regeneration. The authors successfully demonstrated that the studied drug candidate enhanced the migration and differentiation of C2C12 myoblasts via the FAK and PI3K/AKT pathway in vitro and improved myogenesis in injured muscle in mice in vivo. The reported results would pave the way for further investigations aiming at the treatment of muscle loss diseases. The paper could be accepted for publication after addressing the following issues:

Comment 1:

“SCs”, as abbreviation for “satellite cells”, appears for the first time in the introduction L. 48 without indicating what it designates. The authors should mention it earlier L. 41.

Comment 2:

In the presentation 62.82±3.687%, 66.85 ± 3.901%, and 65.13 ± 2.782% the number of decimal places for the percentage error (e.g., 3.687%) is more precise than the value (e.g., 62.82), which typically would be seen as inconsistent in terms of precision. The general rule in reporting results is that the number of decimal places (or significant figures) in the uncertainty (in this case, the percentage error) should match the number of decimal places in the value itself, or at least be rounded to the nearest decimal place based on the precision of the value.

Comment 3:

I am not sure the values reported of the in vivo study presenting the myofiber area and protein expression and obtained from six independent samples can be such close to each other (inside each group). The standard deviations are too low for an in vivo assessment. The authors should double check these results and ensure they are displaying correct SD (and not SEM for example).

Comment 4:

The manuscript contains several orthographic and stylistic mistakes. For instance: “drugs was given” L. 142, “To investigated” L. 236, “di¬erentiation” L. 223, “provid” L. 350. Authors are asked to carefully check the text and correct these mistakes and others.

Author Response

1. Summary

We appreciate you very much for the positive comments and constructive advice on our manuscript entitled “Puerarin promotes the migration and differentiation of my-oblasts by FAK and PI3K/AKT signaling pathways”. (Ref.: biology-3345306). We did do our best to revise our manuscript according to the comments. We have already made a response marked in red font. We would like to express our great appreciation to you.

2. Point-by-point response to Comments and Suggestions for Authors

Comments 1: “SCs”, as abbreviation for “satellite cells”, appears for the first time in the introduction L. 48 without indicating what it designates. The authors should mention it earlier L. 41.

Response 1: Thank you very much for your kind reminder. We sincerely apologize for this oversight and have added the first mention and explanation of 'SCs' in the 41st line of the introduction.

Comments 2: In the presentation 62.82±3.687%, 66.85 ± 3.901%, and 65.13 ± 2.782% the number of decimal places for the percentage error (e.g., 3.687%) is more precise than the value (e.g., 62.82), which typically would be seen as inconsistent in terms of precision. The general rule in reporting results is that the number of decimal places (or significant figures) in the uncertainty (in this case, the percentage error) should match the number of decimal places in the value itself, or at least be rounded to the nearest decimal place based on the precision of the value.

Response 2: Thank you very much for your kind reminder. We have made the corresponding corrections to the inaccuracies in the manuscript (line 175, 176).

Comments 3: I am not sure the values reported of the in vivo study presenting the myofiber area and protein expression and obtained from six independent samples can be such close to each other (inside each group). The standard deviations are too low for an in vivo assessment. The authors should double check these results and ensure they are displaying correct SD (and not SEM for example).

Response 3: Thank you for your attention to the accuracy of the data in the manuscript. We understand your concern regarding the potentially low SD of the muscle fiber area and protein expression values. We confirm that the reported values are the SD, not the SEM. In our study, due to the limited number of samples (six independent samples), we paid special attention to maintaining the consistency of experimental conditions to reduce variability. This strict experimental control helps to minimize differences between individuals, which may lead to the observed lower standard deviations.

Comments 4: The manuscript contains several orthographic and stylistic mistakes. For instance: “drugs was given” L. 142, “To investigated” L. 236, “di¬erentiation” L. 223, “provid” L. 350. Authors are asked to carefully check the text and correct these mistakes and others..

Response 4: Thank you very much for your kind reminder. We sincerely apologize for these oversights and have corrected the corresponding spelling and stylistic errors. We have also thoroughly reviewed the manuscript to ensure that all mistakes have been rectified. We believe these efforts will greatly enhance the quality of our manuscript and meet the standards of the journal. Thank you again for your patience and support.

Reviewer 3 Report

Comments and Suggestions for Authors

The authors study the role of Puerarin on myoblast migration, differentiation and muscle regeneration. The study is interesting and performed well but there are a few concerns with the data presented. Following are the concerns and some suggestions to the authors,

1. The western blot in figure 3A and the quantification in figure 3B do not seem to correlate as the P-FAK levels in the blot don't show a nearly 2-fold increase with Puerarin treatment as indicated by the quantification.

2. In figure 3C, the second lane shows more background in the P-FAK blot and so the quantification data is not so reliable. Also the quantification of P-FAK/FAK in figures 3B and 3D do not correlate for with and without Puerarin treatment as the data is supposed to be exactly same. 

3. Since, Puerarin has an effect on cell migration as well as differentiation based on the data shown, how about looking at the effect of pre-treating the myoblasts before differentiation? It would be good to compare whether the differentiation efficiency is different between pre treatment only or pre+differentiation treatment or differentiation treatment only.

4. In figure 4D, E, the western blot data for MyHC and MyoD in the blot do not correlate well with the quantification data. Based on the blot, there seems to be a higher increase in MyHC levels upon Puerarin treatment and the MyoD levels are not very different but the quantification data seems to indicate otherwise. 

5. In figure 5A and 5C, the P-PI3K blot shows high background so the quantification data is not very reliable. It affects the conclusions made about the Puerarin treatment. 

6. The BaCl2 injury data at D5 post Puerarin treatment looks interesting However, it will be important to perform a time course (D1, D3, D5, D7 and D14 for example) to understand better how the regeneration progresses with and without Puerarin as the model used here is WT mice and so there shouldn't be any defects in muscle regeneration. It is possible that the regeneration is accelerated due to Puerarin treatment and to confirm this a time course study is needed. Also, the authors could test whether initiating the Puerarin treatment before injuring the muscle with BaClcould affect the regeneration further.

Author Response

Response to Reviewer 3 Comments

1. Summary

We appreciate you very much for the positive comments and constructive advice on our manuscript entitled “Puerarin promotes the migration and differentiation of my-oblasts by FAK and PI3K/AKT signaling pathways”. (Ref.: biology-3345306). We did do our best to revise our manuscript according to the comments. We have already made a response marked in red font. We would like to express our great appreciation to you.

2. Point-by-point response to Comments and Suggestions for Authors

Comments 1: The western blot in figure 3A and the quantification in figure 3B do not seem to correlate as the P-FAK levels in the blot don't show a nearly 2-fold increase with Puerarin treatment as indicated by the quantification.

Response 1: Thank you very much for your reminder. The quantitative data reported in Figure 3B are based on the results of three independent experiments. Upon re-examination of the raw data, we identified that in one of the independent experiments, the background of the p-FAK band in the drug treatment group was unusually high, which may have affected the accuracy of the quantitative results. To ensure the reliability of the data, we repeated this experiment and replaced the affected data with the new experimental results. We have updated the experimental results in Figures 3A and 3B. The new Western blot images and corresponding quantitative data are now more consistent and accurately reflect the impact of Puerarin treatment on p-FAK levels. We apologize for the previous inconsistencies in the data and thank you again for pointing out this important issue.

Comments 2: In figure 3C, the second lane shows more background in the P-FAK blot and so the quantification data is not so reliable. Also the quantification of P-FAK/FAK in figures 3B and 3D do not correlate for with and without Puerarin treatment as the data is supposed to be exactly same.

Response 2: Thank you very much for your reminder. To ensure the accuracy of the results, we repeated the experiment in Figure 3C and ensured that the background noise of all lanes was effectively controlled. For the same reason, we have also updated the results of Figure 3A. We believe that through the above measures, our experimental results will be more accurate and reliable, which can better support our scientific discoveries. We apologize for the inconsistency in the previous data and thank you for pointing out this important issue.

Comments 3: Since, Puerarin has an effect on cell migration as well as differentiation based on the data shown, how about looking at the effect of pre-treating the myoblasts before differentiation? It would be good to compare whether the differentiation efficiency is different between pre treatment only or pre+differentiation treatment or differentiation treatment only.

Response 3: Thank you very much for your suggestion. Puerarin is recognized as a stem cell stimulant. In research exploring the role of puerarin in promoting bone or nerve regeneration, it is commonly administered during the cell differentiation phase. Building on this, we concurrently treated myoblasts with puerarin during their migration and differentiation processes. It is an interesting idea to compare the differentiation efficiency between pre-treatment only, pre-treatment+differentiation treatment, or differentiation only treatment. Future studies could delve deeper into the comparative efficacy of these distinct drug administration strategies to optimize therapeutic outcomes.

Comments 4: In figure 4D, E, the western blot data for MyHC and MyoD in the blot do not correlate well with the quantification data. Based on the blot, there seems to be a higher increase in MyHC levels upon Puerarin treatment and the MyoD levels are not very different but the quantification data seems to indicate otherwise.

Response 4: Thank you very much for your question. In the immunoblotting process for MyoD protein, we used GAPDH as a loading control. The imbalance in the loading control affected the expression levels of MyoD protein. We have uploaded the corresponding loading controls for MyoD in the updated figures to ensure that the images and tables appear consistent with each other. Additionally, we have replaced the blots for MyHC and myoG. The quantitative data of the experiment were derived from three independent experiments, and we randomly selected one of them for presentation, which may result in “inconsistencies”. We apologize for any inconvenience caused.

Comments 5: In figure 5A and 5C, the P-PI3K blot shows high background so the quantification data is not very reliable. It affects the conclusions made about the Puerarin treatment.

Response 5: Thank you very much for your reminder. We have re-performed the western blot experiments with special attention to controlling the background noise to ensure the accuracy of the quantitative data, and have updated the relevant results in figure 5A and 5C. Thank you again for your valuable comments.

Comments 6: The BaCl2 injury data at D5 post Puerarin treatment looks interesting However, it will be important to perform a time course (D1, D3, D5, D7 and D14 for example) to understand better how the regeneration progresses with and without Puerarin as the model used here is WT mice and so there shouldn't be any defects in muscle regeneration. It is possible that the regeneration is accelerated due to Puerarin treatment and to confirm this a time course study is needed. Also, the authors could test whether initiating the Puerarin treatment before injuring the muscle with BaCl2 could affect the regeneration further..

Response 6: Thank you very much for your good suggestion. In the BaClâ‚‚-based muscle injury model, histologically, hemorrhagic inflammation is typically observed on day 1, accompanied by significant erythrocyte exudation, and mononuclear cell infiltration peaks on day 3. New muscle fibers of various sizes begin to appear between days 5 and 7. Therefore, we chose day 5 to assess the regenerative effect of puerarin by observing the formation of new muscle fibers. Our findings show that puerarin promotes muscle fiber generation, confirming its potential to accelerate muscle regeneration. Observing histological changes at more time points in the BaClâ‚‚ muscle injury model would provide a more comprehensive evaluation of puerarin's potential impact on muscle regeneration. In fact, we are currently working on this and, if time permits, we will include these additional time points in the study. Furthermore, starting puerarin treatment prior to muscle injury could theoretically allow the drug to reach therapeutic concentrations more rapidly, potentially enhancing its therapeutic effect. We will consider this approach in our future studies. Once again, thank you very much for your valuable comments and suggestions.